# Attribution Analysis of Runoff Variation in Kuye River Basin Based on Three Budyko Methods

**Jiahao Zheng** [1,2], **Yi He** [2], **Xiaohui Jiang** [1,2,*], **Tong Nie** [1,2] and **Yuxin Lei** [2]

1    Shaanxi Key Laboratory of Earth Surface System and Environmental Carrying Capacity, College of Urban and Environmental Science, Northwest University, Xi'an 710127, China; 201920847@stumail.nwu.edu.cn (J.Z.); nietong@stumail.nwu.edu.cn (T.N.)
2    Environmental Engineering, College of Urban and Environmental Science, Northwest University, Xi'an 710127, China; yihe@nwu.edu.cn (Y.H.); leiyuxin@stumail.nwu.edu.cn (Y.L.)
*    Correspondence: xhjiang@nwu.edu.cn

**Abstract:** The Loess Plateau is the main soil erosion area within the Yellow River Basin. Quantifying the contribution rate of climate change and human activities to runoff change can provide support for water resources management in the Yellow River Basin. Kuye River Basin is located in the Loess Plateau. As a first-class tributary of the Yellow River, it was selected as the study area. Runoff from the Kuye River Basin has decreased significantly since the 1990s owing to climate change and anthropogenic coal mining. The main objective of this study was to quantify the contribution and sensitivity of climate change and anthropogenic activities to runoff changes using three popular Budyko and elasticity coefficient methods, as well as to compare the similarities and differences among the three methods. The results show that: (1) Through four mutation point test methods, the change point of runoff in the study period of Kuye River Basin is 1997. (2) The elasticity coefficients calculated by the three Budyko methods showed that during the study period, the runoff was more sensitive to changes in precipitation, followed by the catchment surface characteristic parameters and the potential evapotranspiration. (3) All three Budyko methods can yield reasonable contributions of climate change and human activity to runoff changes. The three methods together indicate that the influence of the catchment surface characteristic parameters is the most important factor for the runoff variation in the Kuye River.

**Keywords:** climate change; attribution analysis; Budyko hypothesis; human activities





## 1. Introduction

Significant changes have occurred in the global climate system. The IPCC [1] report notes that global climate change is a growing problem, with the average global temperature rising by 1.53 °C. Over the previous centuries, climate change has resulted in serious global impacts, including increased rainfall intensity, frequent extreme weather events, and rising sea levels. Climate change also has a serious effect on the water cycle, as an important medium for the exchange of materials and energy in the natural climate, thus attracting significant attention [2,3]. Hydrological processes [4] in river basins have also changed in the context of global climate change [5,6] As a key link in the water cycle, runoff is not only an important pathway between surface water and atmospheric water, but it is also closely related to the development of human society. Moreover, with the continuous societal development and progress, changes in the conditions of basin substrates also affect regional hydrological processes. The joint interference of both factors yields a highly complex runoff process. Therefore, a comprehensive understanding of the causes of runoff changes [7] in watersheds and the mechanisms of their hydrothermal balance is necessary to provide an effective scientific basis and potential adaptation policies for watershed and land resource management [8,9].

Attribution analyses of runoff changes are a debated issue in the current hydrological research. Climate change and human activities are generally thought to be the main factors that influence changes in runoff; the core issue of runoff attribution lies in the distinction between the effects of climate change and human activities. The methods used in recent runoff attribution studies can be divided into three main categories: general statistical models, coupled hydrothermal models, and hydrological models. Among general statistical models, double-mass curves (DMCs) are a common method for the analysis of the evolution of hydro-meteorological elements [10], which are useful for comparative analyses. DMCs are characterized by low data requirements and high transferability, thus rendering them more practical than water balance equations and hydrological models in hydrologic benefit evaluations [11]. Jin et al. [12] used the DMC method to detect the contribution of climate change and human activities to runoff reductions of −20 and 120%, respectively, while delineating the specific contribution of human activities. A Pirnia et al. [13] used the DMC method to analyze the runoff variability in the Tajan River Basin, Iran, which showed that the contributions of climate change and human activities to the predicted runoff reduction were 24.68 and 75.32% for the dry season climate contribution of −30.68%, while human activity was 130.68% during the same period. Essentially, the DMC method uses mathematical and statistical models to attribute the runoff, allowing distinction among subfactors; the lack of physical mechanisms in the DMC method itself makes its application relatively limited. Hydrologic models are the most widely used for runoff change attribution analysis. There are two types of hydrologic models: one with a simple structure and parameters that lacks physical meaning, known as the lumped hydrologic model [14], and the distributed hydrologic model [15], which accounts for this shortcoming; its parameters represent the climate characteristics and subsurface conditions of a basin, which can accurately describe the hydrologic processes. The Soil and Water Assessment Tool [16] (SWAT) model can accurately reflect the spatial variability of complex hydrologic processes in watersheds; previous studies have used SWAT models to attribute runoff variability in different watersheds [17–19] The SWAT model has a strong physical mechanism, which can accurately reflect watershed production and sink processes and provide a deeper understand of the causes of runoff changes. However, this model requires a large amount of data and the model database requires an extended development period. The coupled hydrothermal model based on the Budyko hypothesis, which accounts for the water balance, energy balance, basin substrate conditions, and interaction of various factors within the basin, has become a tool for investigating the complex relationships among the hydrological elements of a basin, which has been widely used in recent years for attribution analyses of runoff changes in different basins [20,21]. Li et al. [22] used the Budyko framework to analyze the attribution of runoff changes in the main tributaries along the middle reaches of the Yellow River, China, as well as exploring the spatial and temporal distribution characteristics of the impact that human activities have on runoff. Liu et al. [23] applied the Choudhury–Yang equation to calculate the contribution to runoff variability in the Lancang River Basin, concluding that precipitation variability is the main cause of runoff variability. Meanwhile, many empirical formulas have been derived based on the Budyko hypothesis; most studies have selected the Choudhury and Yang equation. Fewer studies compare the consistency and uncertainty in the quantitative results calculated by different Budyko methods [24]. In this paper, we selected three widely used Budyko methods, as well as the elasticity coefficient method, to investigate runoff changes in the Kuye River Basin. The similarities and differences between the three methods were compared while deriving the factors that affect runoff changes and the sensitivity of runoff to the influencing factors.

The hydrological data of Wenjiachuan hydrological station show the annual variation in the measured runoff and average precipitation at the watershed surface from 1956 to 2018 at the Kuye River hydrological station, which showed a significant decreasing trend in the annual runoff, but no significant decreasing trend in the annual precipitation. From this, we speculated that runoff changes in the Kuye River Basin may mainly influenced by human activities. Yang et al. [25] used the Choudhury and Yang equation to obtain

the impact of climate change and human activities on runoff change in the Yellow River Basin from 1961 to 2010; therefore, this study aims to quantify the effects that climate and anthropogenic changes have on runoff in a longer time series and compare the differences between the three methods using runoff observation data, combining trend analysis, and the three Budyko methods. First, we analyzed the hydrometeorological data using a Mann–Kendall trend test and multiple change point test. We then applied the three Budyko methods to calculate the elasticity coefficients of climate and human activities with respect to runoff changes, finally deriving the contribution of climate change and human activities to changes in the runoff. This study provides a reference for the selection of different Budyko methods in terms of runoff change attribute analyses in similar watersheds, as well as a scientific basis for water resource utilization and land management in Shaanxi.

## 2. Study Area and Data

### 2.1. Study Area

The study area is the Kuye River Basin, a first-class tributary of the Yellow River, located upstream of the Wenjiachuan hydrological station, between 108°28′ E–110°45′ E and 38°22′ N–39°50′ N. This basin is located in the middle reaches of the Yellow River, originating in the Inner Mongolia Autonomous Region, passing through the Ijinholo Banner territory and Fugu County in Shaanxi Province, and flowing into the Yellow River in Shenmu County, China. The total length of the river is 242 km, with a basin area of 8706 km$^2$. The Kuye River Basin has an arid to semi-arid continental climate with notable seasonal changes in precipitation. The spatial and temporal distribution of precipitation is highly uneven: June to September precipitation accounts for 75–81% of the annual precipitation, where the precipitation in July and August accounts for 50–60% of the annual precipitation, mainly as heavy rainfall. Spatially, precipitation is less in the north and west and greater in the east. The average multi-year runoff in the Kuye River Basin is 5.042 × 10$^8$ m$^3$, the average annual precipitation ranges from 161.63 to 681.76 mm, and the average annual temperature ranges from 6.06 to 9.34 °C. Figure 1 shows the geographical location of the Kuye River Basin and the spatial distribution of the meteorological and hydrological stations.

### 2.2. Data Sources

The observed daily runoff data for 1956–2018 from Wenjiachuan station were obtained from the Yellow River Conservancy Commission (http://www.yrcc.gov.cn/) (accessed on 1 October 2021). Annual runoff was calculated based on the watershed area. Daily meteorological data from five national meteorological stations for 1956–2018 were obtained from the National Meteorological Center (http://data.cma.gov.cn) (accessed on 1 October 2021), including precipitation, temperature (mean, maximum and minimum), sunshine hours, and mean wind speed, which were interpolated using the distance direction weighting method to obtain watershed-scale meteorological data. Potential evapotranspiration was calculated using the FAO-modified Penman–Monteith formula:

$$\text{ET}_0 = \frac{0.408(R_n - G) + \gamma \frac{900}{273+T} U_2 (e_S - e_a)}{\Delta + \gamma(1 + 0.34U_2)} \tag{1}$$

where $\text{ET}_0$ is the reference evapotranspiration (mm day$^{-1}$), Rn is the net radiation at the crop surface (MJ m$^{-2}$ day$^{-1}$), G is the soil heat flux density (MJ m$^{-2}$ day$^{-1}$), T is the mean daily air temperature at a height of 2 m (°C), $U_2$ is the wind speed at a height of 2 m (m s$^{-1}$), es is the saturation vapor pressure (kPa), ea is actual vapor pressure (kPa), es−ea is the saturation vapor pressure deficit (kPa), $\Delta$ is slope vapor pressure curve (kPa °C$^{-1}$), and $\gamma$ is the psychrometric constant (kPa °C$^{-1}$).

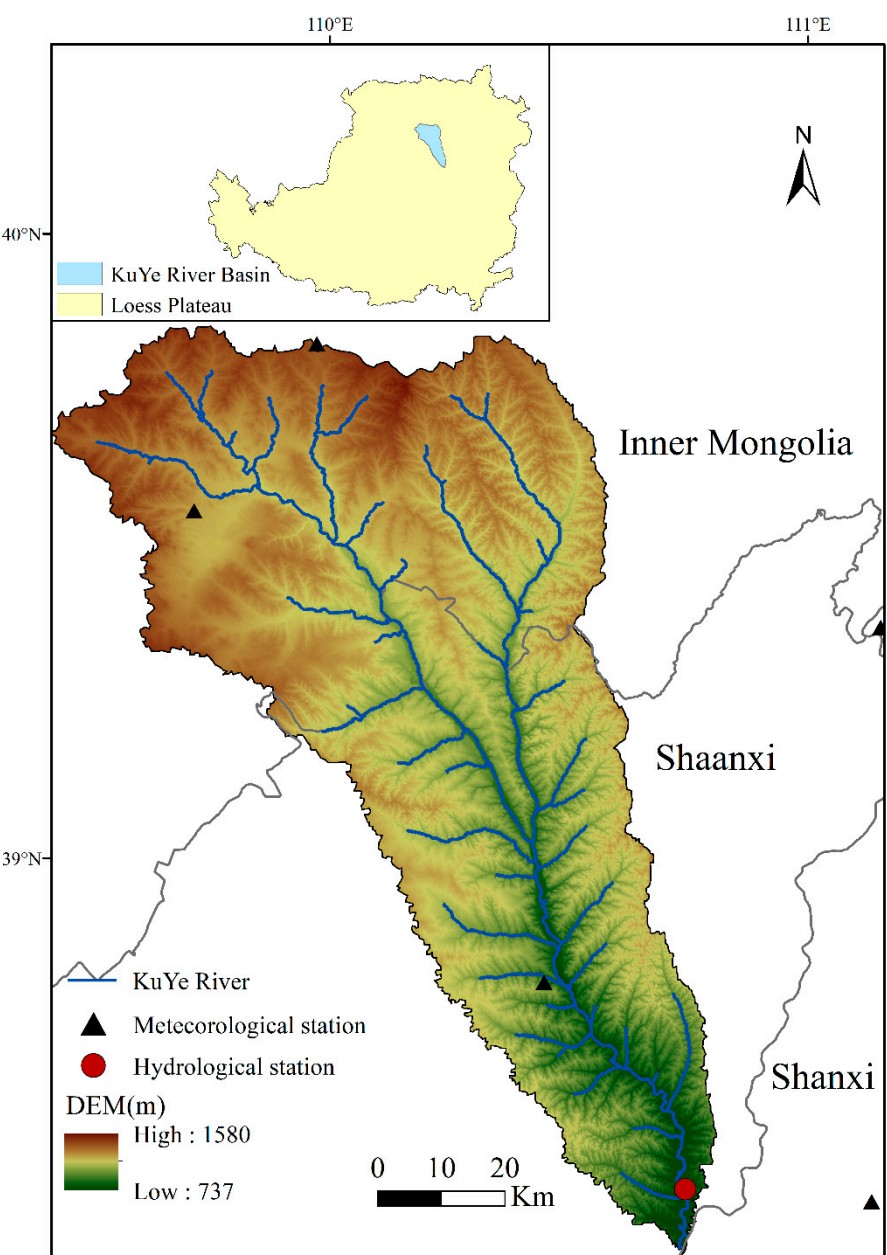

**Figure 1.** Topographical map of the Kuye River Basin and distribution of the meteorological and hydrological stations.

The Normalized difference vegetation index (NDVI) data from the Grotto River Basin were obtained using the "Long Time-Series Chinese Vegetation Index Dataset–GIMMS NDVI" from the Western China Environmental and Ecological Science Data Center. The GIMMS NDVI data are monthly data at a spatial resolution 8 km from 1981 to 2015. The annual average NDVI values of the Kuye River Basin were calculated based on this dataset as the basic data for analyzing the 30-year substrate preparation changes in the basin. Land use data were used as a source to analyze the changes in land use that occurred in 1980 and 2018 using Landsat remote sensing image data from U.S. Landsat.

### 3. Methodology

*3.1. Trend Test and Mutation Analysis Methods*

3.1.1. Mann–Kendall Trend Analysis

The Mann–Kendall trend test, which is a widely used statistical method recommended by the World Meteorological Organization, can effectively determine the significance of the trend of a natural process [26,27]. In this study, the trends in the runoff depth, precipitation and potential evapotranspiration were obtained via Mann–Kendall trend analysis, which is a simple calculation process with intuitive and accurate results.

$$S = \sum_{j=1}^{n-1} \sum_{i=j+1}^{n} \text{sgn}(x_i - x_j) \tag{2}$$

where

$$\text{sgn}(x_i - x_j) = \begin{cases} 1 & x_i > x_j \\ 0 & x_i = x_j \\ -1 & x_i < x_j \end{cases} \tag{3}$$

where

$$\text{Var}(S) = n(n-1)(2n+5)/18 \tag{4}$$

$$Z = \begin{cases} (S-1)/\sqrt{\text{Var}(S)} & S > 0 \\ 0 & S = 0 \\ (S-1)/\sqrt{\text{Var}(S)} & S < 0 \end{cases} \tag{5}$$

where $X_i$ and $X_j$ are the values corresponding to years i and j in the time series, n is the length of the time series data, and Z is the trend of time series data; if $Z > 0$, the time series data shows an increasing trend over time. Otherwise, the time series data show a decreasing trend over time. When $|Z| > Z_{(1-\alpha/2)}$, the null hypothesis is rejected and the time series data show a significant trend. $Z_{(1-\alpha/2)}$ denotes the area to the right of the density function quantile, which uses the lower quantile. The value of $Z_{(1-\alpha/2)}$ can be obtained from the normal distribution table, and the corresponding value of $Z_{(1-\alpha/2)}$ is 1.96 at a significance level of $\alpha = 5\%$ [28].

3.1.2. Mutation Analysis Methods

Due to both climatic variability and human activities, the hydrological series has a significant variation in its statistical pattern around a certain point in time, which is defined as the variation point of the series. There is significant uncertainty associated with the testing of mutation points, such that it is necessary to accurately obtain the mutation points of runoff occurrence during the study period. In this study, the Pettitt test method [29], cumulative distance level method, moving *t*-test method, and Yamamoto method [30] were used to detect mutation points in the runoff depth data in the Kuye River Basin, Pettitt test method [31] is a non-parametric test that assumes the existence of a trend in the series and determines the time of mutation by examining the time of change in the mean value of the time series. The first step is to find the first-level mutation point in the entire time series; the original series is then divided into two sequences to continue to detect the new mutation point, such that there may be more than one mutation point. Finally, the mutation point should be obtained according to the specific cause analysis, which determines the mutation by testing the significance of the difference between the means of two random samples. For the time series, we first artificially set a base year, divided the series into two subsequences, and calculated the mutation index S/N (i.e., the signal-to-noise ratio) of the base year, which is defined as a mutation when S/N > 1.0, and a strong mutation when S/N > 2.0. The cumulative distance level method is a mean–based test, which can determine the degree of data point dispersions and long-term trends and mutation times in the time series by observing the difference product curve. The results of the four methods were then combined to divide the time series of the runoff depth into two periods: the base

period (assumed to be the natural condition without human activities) and the impacted period (human activities).

*3.2. Identification of Runoff Change Attribution*

3.2.1. Budyko Hypothesis

The hydrological elements of the basin across a long-term scale follow the water balance equation, expressed as follows:

$$R = P - ET + \Delta S \tag{6}$$

where R is the mean annual runoff, P is the mean annual precipitation, ET is the mean annual actual evapotranspiration, and $\Delta S$ is the change in the water storage change in a basin, which approaches zero over an extended long period.

Budyko [32] suggested that the balance between the atmospheric water supply to the land surface and atmospheric evaporation demand determines long-term average evapotranspiration (ET) from the land surface, which led to the Budyko hypothesis. The moisture supply can be expressed in terms of precipitation, while the net radiation or potential evapotranspiration represents the atmospheric evaporative demand, thus leading to the general form of the Budyko hypothesis:

$$\frac{ET}{P} = f\left(\frac{ET_0}{P}\right) = f(\Phi) \tag{7}$$

where $ET_0$ is the potential evapotranspiration, P is the precipitation, and $\Phi$ is the dryness, which is the ratio of potential evapotranspiration to precipitation.

The initial Budyko hypothesis [33] did not consider certain characteristics, such as subsurface and watershed areas. Based on this, several studies have proposed a series of Budyko empirical formulas that reflect the subsurface factors in a single parameter [34,35] and their theoretical derivations and verifications have been subsequently carried out. Among these forms, the equations of Fu, Yang and Choudhury, and Wang and Tang (Table 1) were selected in this study for attribution identifications of the runoff changes in the Kuye River from 1956 to 2018.

**Table 1.** Three Budyko-type equations for estimating actual evapotranspiration employed in this study.

| Formula | Parameter | References |
|---|---|---|
| $ET/P = [1 + ET_0/P - (1 + (ET_0/P)^{\omega})]^{1/\omega}$ | $\omega$ | Fu [24,36,37] |
| $ET/P = 1/[1 + (P/ET_0)^n]^{1/n}$ | $n$ | Yang and Choudhury [38,39] |
| $ET/P = \dfrac{1 + ET_0/P - \sqrt{(1 + ET_0/P)^2 - 4\varepsilon(2-\varepsilon)ET_0/P}}{2\varepsilon(2-\varepsilon)}$ | $\varepsilon$ | WANG and TANG [40] |

Notes: $\omega$ and n are the hydrothermal coupling control parameters and $\varepsilon$ is the ratio of the initial evaporation to the total evaporation (later expressed uniformly as ni). These parameters can be obtained from the equations in Table 1, combined with Equation (6).

Based on the equations in Table 1 combined with the water balance equation, we can obtain the following:

$$R = P - \left(P + ET_0 - P\left(1 + \left(\frac{ET_0}{P}\right)^{n_{FU}}\right)^{\frac{1}{n_{FU}}}\right) \tag{8}$$

$$R = P - \frac{P \times ET_0}{(P^{n_{CY}} + ET_0^{n_{CY}})^{\frac{1}{n_{CY}}}} \tag{9}$$

$$R = P - \frac{P + ET_0 - P \times \sqrt{\left(1 + \frac{ET_0}{P}\right)^2 - 4n_{WT}(2 - n_{WT})\frac{ET_0}{P}}}{2n_{WT}(2 - n_{WT})} \tag{10}$$

### 3.2.2. Elasticity Coefficient

Scchaake [41] derived an equation for the sensitivity of runoff to climate variables based on the Budyko curve, introducing the climate elasticity coefficient, which is defined as the ratio in the rate of change in the runoff to the rate of change in a climate factor. Subsequently, several studies introduced multiple elements to establish a multi-parameter climate elasticity coefficient model [42], which can be expressed as follows:

$$\varepsilon_x = \frac{\partial R/R}{\partial x_i/x_i} \tag{11}$$

where $\varepsilon_x$ is the elasticity coefficient of the runoff to the independent variables and $x_i$ represents P, $ET_0$ and n.

The elasticity coefficients corresponding to the three Budyko-based methods can be derived from Equations (8)–(10).

### 3.2.3. Quantifying Contributions of Changes in the Climate and Subsurface Factors to Runoff Changes

Based on the abrupt change point test, the changes in runoff depth, precipitation, potential evapotranspiration, and subsurface parameters from the base period to the anthropogenic period were expressed as follows:

$$\Delta R = R_2 - R_1 \tag{12}$$

$$\Delta P = P_2 - P_1 \tag{13}$$

$$\Delta ET_0 = ET_{0_2} - ET_{0_1} \tag{14}$$

$$\Delta n_i = n_{i_2} - n_{i_1} \tag{15}$$

where $\Delta R$, $\Delta P$, $\Delta ET_0$, and $\Delta n_i$ are the magnitudes of change from the base period to the anthropogenic period.

Assuming that P, $ET_0$, and $n_i$ in Equations (8)–(10) are independent variables, Equations (8)–(10) can be expressed as R = f (P, $ET_0$, $n_i$). We can then obtain the total differential of R, which can be written as follows:

$$dR' = dR_P + dR_{ET_0} + dR_{n_i} = \frac{\partial f}{\partial P}dP + \frac{\partial f}{\partial ET_0}dET_0 + \frac{\partial f}{\partial n_i}dn_i \tag{16}$$

where $dR'$ is the total value of the change in the runoff change caused by P, $ET_0$, and n, and dRX is the change in the runoff change caused by P, $ET_0$ and $n_i$.

Therefore, the contribution of climate change (P, $ET_0$) and the subsurface, $n_i$, to runoff changes can be calculated as follows:

$$\eta_x = \frac{dR_x}{dR} \times 100\% \tag{17}$$

where $\eta_x$ is the contribution of the runoff change in P, $ET_0$, and $n_i$.

The relative error (RE) was adopted to quantify the performance of the elasticity coefficient method, which was defined as follows:

$$RE = \frac{R_{cal} - R_{obs}}{R_{obs}} \times 100\% \tag{18}$$

where $R_{cal}$ and $R_{obs}$ are the values of the calculated and observed runoff depths, respectively.

## 4. Results and Analysis

### 4.1. Trend Analysis of Hydrometeorological Variables

Our analysis of the variability in the hydroclimatic data in the Kuye River Basin during the study period is the basis of this study and can also partially reflect the reasonableness of the study results. Figure 2 shows the results of the Mann–Kendall trend test. Potential evapotranspiration showed a non-significant increasing trend in the first 23 years and a significant decrease from 1984 to 2005. The runoff depth showed a non-significant increase from 1956 to 1982 and a significant decrease from 1996 onward. Figure 3 shows the interannual trends in the precipitation, potential evapotranspiration, and runoff depth in the Kuye River Basin from 1956 to 2018, where precipitation shows a fluctuating increasing trend during the study period, whereas the potential evapotranspiration and runoff show a decreasing trend.

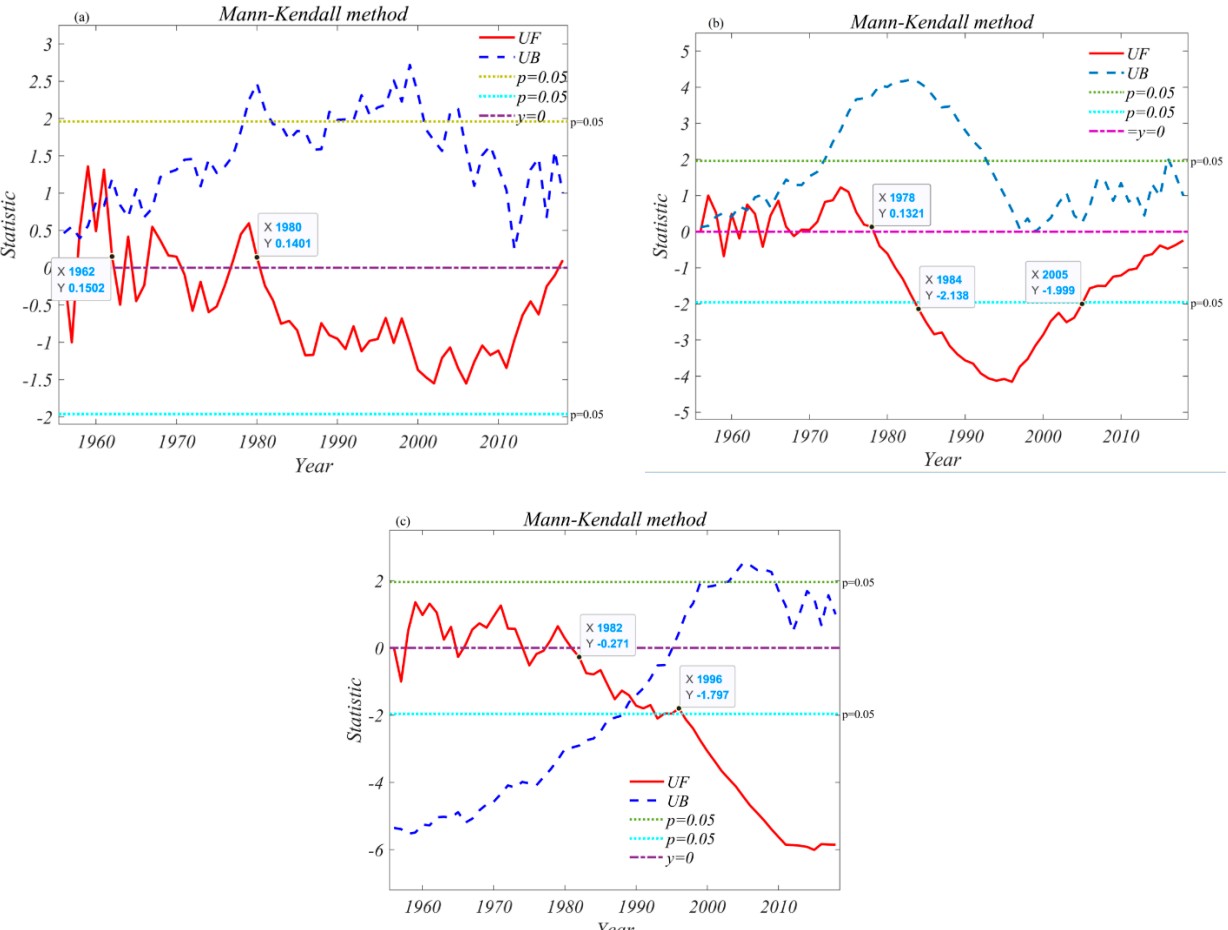

**Figure 2.** Mann–Kendall (MK) trend test of the (**a**) annual precipitation, (**b**) potential evapotranspiration, (**c**) runoff depth. UF, UB, and p are the statistics in the Mann–Kendall trend test, respectively.

Based on our results, we can conclude that the precipitation, potential evapotranspiration, and runoff depth do not have a significant correlation; the decrease in the runoff depth was more significant compared with the decrease in precipitation.

### 4.2. Change Point Analysis of Runoff

To obtain accurate runoff change points, we used four methods (i.e., the Pettitt test method, cumulative distance level method, moving *t*-test method and Yamamoto method) to examine the runoff depth during the study period, as shown in Figure 4. For the Pettitt test method, the change point for the runoff occurred in 1997. The results of the cumulative

distance level method, the Yamamoto method and the moving T method are all for 1996 and 1998.

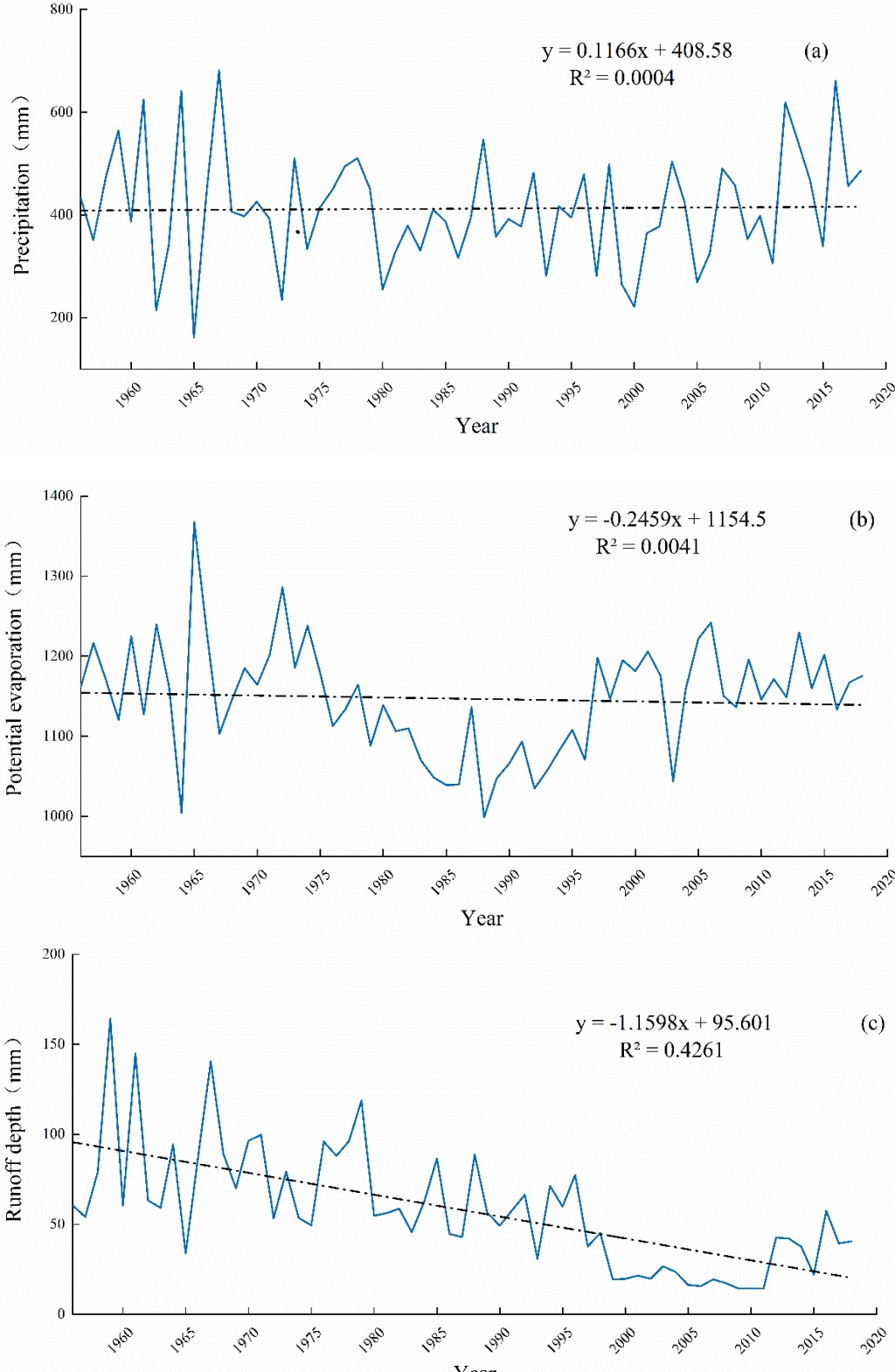

**Figure 3.** Trends in (**a**) the annual precipitation, (**b**) potential evapotranspiration, and (**c**) runoff depth from 1956 to 2018 in the Kuye River Basin.

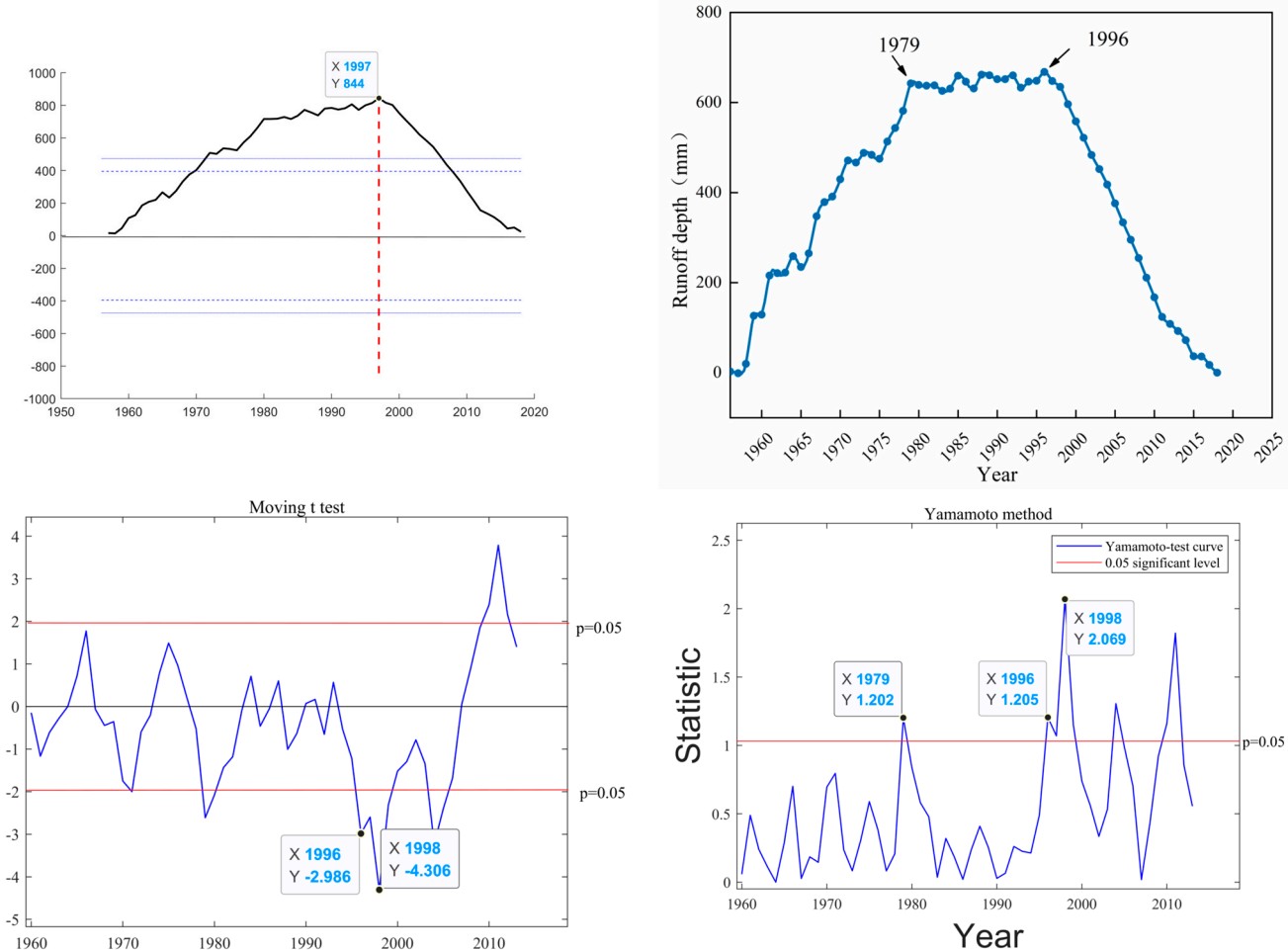

**Figure 4.** Pettitt test method, cumulative distance level method, moving *t*-test method and Yamamoto method.

Thus, by synthesizing the results of the above four methods, we used 1997 as the change point of the runoff, which is consistent with the results reported by Song et al. [43]. This result is consistent with the timing of the massive coal mining in the Kuye River Basin and the implementation of the Chinese government's "return of farmland to forest" policy. This means that from 1997 to the present, a large number of anthropogenic activities have affected the natural ecological processes in the Kuye River Basin. The study period was classified as the base period from 1956 to 1996 and the anthropogenic impact period from 1997 to 2018. Table 2 lists the characteristic values of the hydroclimatic variables for each period. Based on the hydroclimatic characteristics of the two different periods and the catchment surface characteristics corresponding to the three Budyko methods, the relative rate of change in the runoff depth compared to the base period decreased by 210.2%, whereas the precipitation and potential evapotranspiration showed a slight increase. In contrast, the catchment surface characteristic parameters increased by 25.14, 34.05, and 54.88% in the three Budyko methods. Additionally, there was no significant change in the drought index compared with the base period, indicating that there was no significant difference in the degree of warmth or wetness between the two periods. We can therefore roughly conclude that the main influence on the change in runoff was the change in the surface characteristics of the basin.

**Table 2.** Meteorological data and the parameters (n) of three Budyko formula for two periods.

| Period | P/mm | $ET_0$/mm | R/mm | nFU | nCY | nWT | R/P | $ET_0$/P |
|---|---|---|---|---|---|---|---|---|
| 1956–1996 | 411.414 | 1133.043 | 74.197 | 1.983 | 1.249 | 0.279 | 0.180 | 2.752 |
| 1997–2018 | 413.986 | 1171.982 | 27.554 | 2.649 | 1.894 | 0.619 | 0.076 | 2.830 |
| Relative change (%) | 0.06 | 3.33 | −210.20 | 25.14 | 34.05 | 54.88 | −170.9 | 2.71 |

*4.3. Attribution analysis of Kuye River Runoff Changes*

4.3.1. Sensitivity Analysis of Runoff Depth to Changes in Climatic Factors and Catchment Surface Characteristic Parameters

From the elasticity coefficients calculated via the three Budyko methods in Table 3, the potential evapotranspiration and catchment surface characteristic parameters are negatively correlated with the runoff, while precipitation is positively correlated with the runoff. During the base period, the Wang and Tang equation yielded that for each 1% increase in the potential evapotranspiration or n parameter, n decreased the runoff depth by 0.929 or 0.506%, whereas for each 1% increase in the precipitation, the runoff depth increased by 1.928%. For each 1% increase in the precipitation, potential evapotranspiration, or n parameter during the anthropogenic impact period, the runoff depth increases by 2.267%, decrease by 1.267%, or 2.762%. The equation in Fu showed that for each 1% increase in the precipitation, potential evapotranspiration, or n parameter during the base period, the runoff depth increased by 1.924%, decrease by 0.923 or 2.98%. For each 1% increase in the precipitation, potential evapotranspiration, or n parameter during the anthropogenic impact period, the runoff depth will increase by 2.6% or decreased by 1.6 and 3.72%. Choudhury and Yang showed that for each 1% increase in the precipitation, potential evapotranspiration or n parameter during the baseline period, the runoff depth increased by 2% or decreased by 1 and 1.917%. For each 1% increase in the precipitation, potential evapotranspiration or n parameter during the anthropogenic impact period, the runoff depth increased by 2.72% or decreased by 1.715 or 2.72%. Moreover, the absolute value of the elasticity coefficient of precipitation was the largest based on the Wang and Tang and Choudhury and Yang equations during the base period while the n parameter is the largest for the Fu equation. Within the anthropogenic impact period, the elasticity coefficients of the n parameter are the largest for the three Budyko methods, indicating that the depth of runoff in the Kuye River Basin was increasingly sensitive to the elasticity coefficients, regardless of the method. Overall, the values of all of the elasticity coefficients increased, indicating that the watershed became more sensitive to these three factors and less sensitive to other factors that were not considered to have an impact on the runoff depth in this study.

**Table 3.** Sensitivity of R to P, $ET_0$ and n for the three Budyko methods 1956–1996 (base period) and 1997–2018 (anthropogenic impact period) in the Kuye River.

| Period | Elasticity Coefficient | | | | | | | | |
|---|---|---|---|---|---|---|---|---|---|
| | Wang & Tang | | | Fu | | | Choudhury & Yang | | |
| | $\varepsilon_P$ | $\varepsilon_{ET_0}$ | $\varepsilon_n$ | $\varepsilon_P$ | $\varepsilon_{ET_0}$ | $\varepsilon_n$ | $\varepsilon_P$ | $\varepsilon_{ET_0}$ | $\varepsilon_n$ |
| 1956–1966 | 1.928 | −0.929 | −0.506 | 1.924 | −0.923 | −2.98 | 2.00 | −1.00 | −1.917 |
| 1997–2018 | 2.267 | −1.267 | −2.762 | 2.600 | −1.600 | −3.72 | 2.72 | −1.715 | −2.750 |

Figures 5–7, show the trends in the elasticity coefficients for the precipitation, potential evapotranspiration and n parameter from 1956 to 2018, respectively. For the precipitation elasticity coefficient, the sensitivity of the runoff depth from the Choudhury and Yang equation was the highest during the baseline period, while the sensitivity of the Wang and Tang equation was higher in the anthropogenic impact period. The elasticity coefficients of the three methods were comparable for the potential evapotranspiration elasticity coefficient

and n parameter; the absolute values for the elasticity coefficients for the three methods fluctuated and increased within the anthropogenic impact period. Generally, the trends in the three methods were consistent, and the absolute values of the elasticity coefficients fluctuated and increased during the study period, indicating the increased sensitivity of the runoff depth to these three factors. Compared with the other two equations, the Wang and Tang equations for the three elasticity coefficients yielded increased fluctuations.

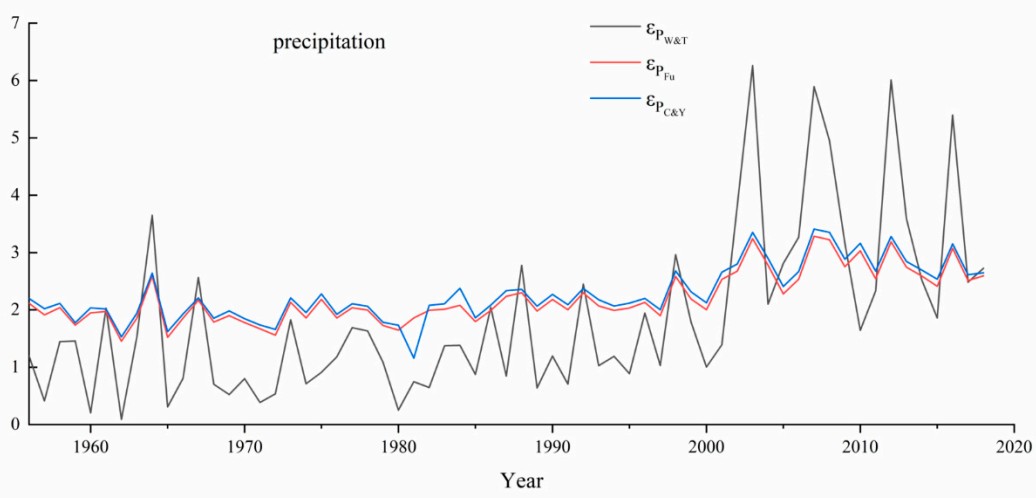

**Figure 5.** Elasticity coefficients of precipitation for three Budyko methods from 1956 to2018.

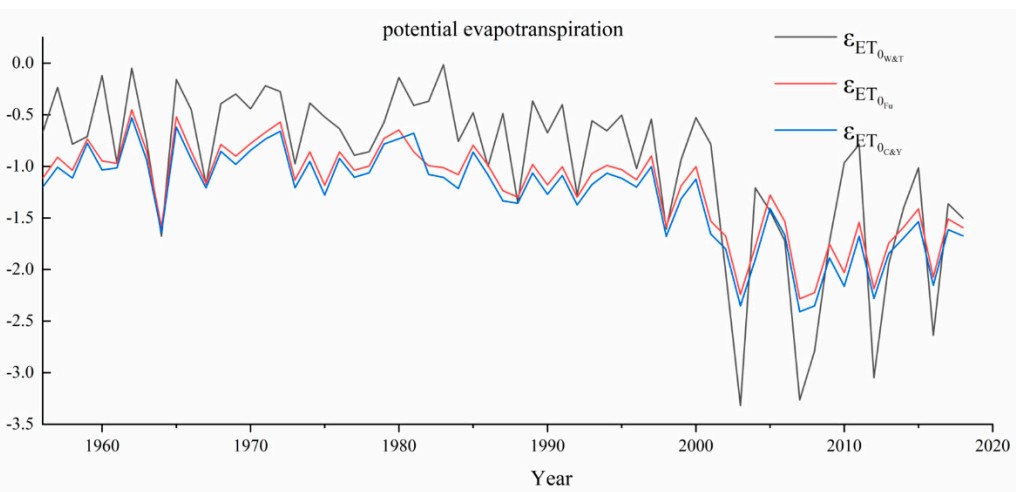

**Figure 6.** Elasticity coefficients of the potential evapotranspiration for the three Budyko methods from 1956 to 2018.

### 4.3.2. Attribution Identification of Runoff Change

The contributions of P, $ET_0$, and n to the change in the runoff depth was calculated using the three Budyko methods based on Equation (17). From Tables 4–6, the Wang and Tang [40] equation showed that the contribution of precipitation-induced increases in the runoff depth was approximately 0.73 mm, as compared with the baseline period, and the contribution of the potential evapotranspiration change to the decrease in the runoff depth of 2.86 mm was 5.08%; however, changes in the n parameter led to a decrease in the runoff depth of 54.19 mm, where the contribution of the change in the n parameter to the change in the runoff depth was 96.22%. The other two equations also obtained similar results. The equation in Fu [36,37] showed that during the anthropogenic impact period, the precipitation variation increased the runoff depth by 0.75 mm with a contribution of 1.26% while the potential evapotranspiration and n parameter variations decreased the runoff depth by 2.17 and 58.29 mm with a contribution of 3.63 and 97.64%, respectively. The Choudhury

and Yang [38,39] equation yielded increments in the runoff depth due to precipitation of 0.78 mm. The reduction in the runoff depth due to potential evapotranspiration was approximately 2.34 mm and the reduction in the runoff depth due to the n parameter was 59.81 mm. The contributions of these three factors were 1, 3, and 97%.

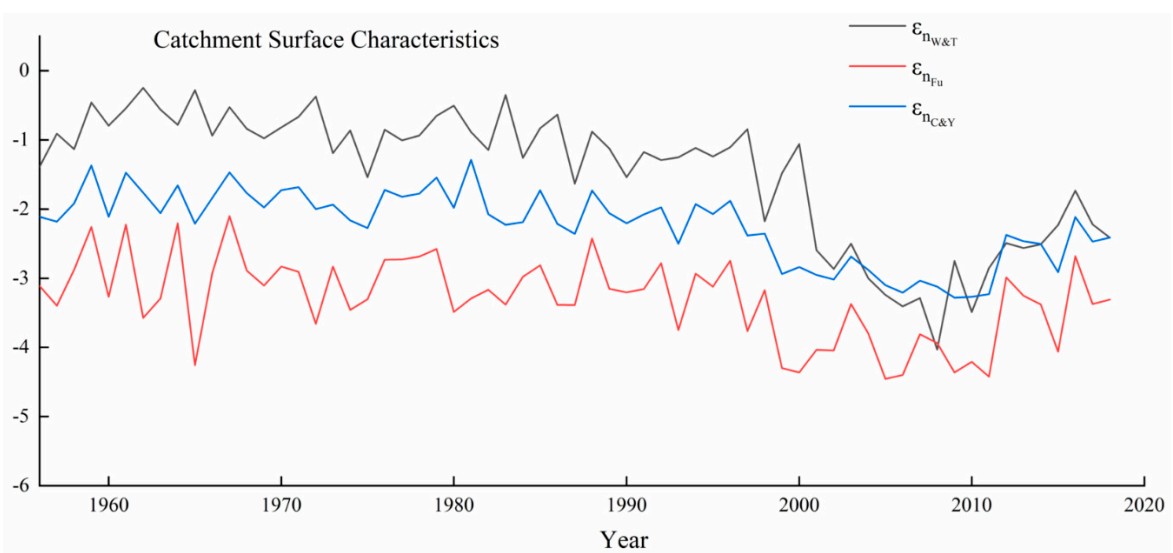

**Figure 7.** Elasticity coefficients for the n parameter for the three Budyko methods from 1956 to 2018.

**Table 4.** Contributions of P, ET0 and n to runoff depth change by Wang and Tang equation.

| Base Period | Impacted Period | $dR_P$ | $dR_{ET_0}$ | $dR_n$ | $dR$ | $dR'$ | RE (%) | Contribution Ratio (%) | | |
| --- | --- | --- | --- | --- | --- | --- | --- | --- | --- | --- |
| | | | | | | | | $\eta_P$ | $\eta_{ET_0}$ | $\eta_n$ |
| 1956–1996 | 1997–2018 | 0.73 | −2.86 | −54.19 | 57.91 | 56.32 | −2.7 | 1.30 | −5.08 | −96.22 |

Note: The table "−" indicates decrease.

**Table 5.** Contributions of P, E0 and n to runoff depth change by Fu equation.

| Base Period | Impacted Period | $dR_P$ | $dR_{ET_0}$ | $dR_n$ | $dR$ | $dR'$ | RE (%) | Contribution Ratio (%) | | |
| --- | --- | --- | --- | --- | --- | --- | --- | --- | --- | --- |
| | | | | | | | | $\eta_P$ | $\eta_{ET_0}$ | $\eta_n$ |
| 1956–1996 | 1997–2018 | 0.75 | −2.17 | −58.29 | 57.91 | 59.70 | −3.01 | 1.26 | −3.63 | −97.64 |

Note: The table "−" indicates decrease.

**Table 6.** Contributions of P, E0 and n to runoff depth change by Choudhury and Yang equation.

| Base Period | Impacted Period | $dR_P$ | $dR_{ET_0}$ | $dR_n$ | $dR$ | $dR'$ | RE (%) | Contribution Ratio (%) | | |
| --- | --- | --- | --- | --- | --- | --- | --- | --- | --- | --- |
| | | | | | | | | $\eta_P$ | $\eta_{ET_0}$ | $\eta_n$ |
| 1956–1996 | 1997–2018 | 0.78 | −2.34 | −59.78 | 57.91 | 61.34 | 5.59 | 1.28 | −3.81 | −97.47 |

Note: The table "−" indicates decrease.

The relative errors between the difference in the runoff depths calculated by the three methods and the change values for the actual runoff depths in the two periods were 2.7, 3.01, and 5.59%. This shows that the methods used and the results obtained in this study can reasonably assess the effects that climate change and human activities have on changes in the runoff.

## 5. Discussion and Conclusions

### 5.1. Discussion

The results of the three Budyko methods are similar in terms of their contribution to the runoff variability, which indicates that the three methods are well suited to the Kuye River Basin. From the analysis of the influence that the elasticity coefficients of the climatic factors and surface characteristic parameters have on runoff changes, the trends and values from the Fu and Choudhury and Yang equations are similar, mainly because the n parameter in both equations represents the hydrothermal coupling control parameter, which combines many factors, such as the soil and vegetation in a watershed. The n parameter in the Wang and Tang equation is defined as the ratio between the initial evaporation and total evaporation, which refers to the evaporation of the vegetation and topsoil; as the soil conditions in the Kuye River Basin do not vary significantly, the results are similar. We suggest that the strong fluctuation in the n parameter in the Wang and Tang equation, starting around 2000, is related to the significant increase in the NDVI values beginning in the year characterized by an abrupt change, as shown in Figure 8b.

In the Budyko hypothesis, runoff is mainly influenced by meteorological factors (precipitation, potential evapotranspiration) and surface characteristic parameters. Among the subsurface factors, land use and NDVI play a significant role. Yan et al. [44] analyzed the NDVI value as a characteristic parameter of the lower bedding surface based on land use directly and in the attribution identification of runoff changes in the source area of the Yellow River. According to Table 7, the 1980–2018 land use transfer matrix, it can be seen that for the Kuye River Basin land use change is greater for construction land and grassland, among which 160.36 km$^2$ and 549.68 km$^2$ of cropland and grassland evolved into construction land, while only 4.47 km$^2$ and 29.25 km$^2$ of construction land were converted into cropland and grassland during the same period. The massive coal mining in the northern part of the Kuye River Basin, which started in the 1990s, became one of the important factors in the anthropogenic alteration of the substratum in the basin; Jin et al. [45] used the SIMHYD–PML hydrological model to assess the contribution of coal mining to the reduction in runoff of up to 59%, and the large amount of fallowing and reforestation of the Kuye River Basin started after 2000 for policy reasons, which also led to a good improvement in NDVI. Figure 8a shows the change of vegetation cover in the basin between 1980 and 2018 (b) shows the NDVI trend fit [46]. The changes in land use and vegetation cover jointly determine the values of Catchment surface characteristic parameters in the Budyko method. However, there are still some uncertainties in this study: observational uncertainty in meteorological station data; interpolation of precipitation and potential evapotranspiration data for the study area may deviate from the actual distribution. Furthermore, the presence of many silt dams or man-made water extraction projects in the Kuye River Basin can also be a factor in the assessment error, and this part of water should be included in the influence in future studies.

**Table 7.** Land use transfer matrix for 1980–2018.

| Land Use Type (km$^2$) | Construction Land | Cropland | Forest | Grassland | Unused Land | Water Area | 2018 Total |
|---|---|---|---|---|---|---|---|
| Converted to Construction land | 25.93 | 160.36 | 42.16 | 549.68 | 48.83 | 35.15 | 862.11 |
| Converted to Cropland | 4.47 | 492.55 | 16.11 | 670.05 | 56.21 | 24.50 | 1263.90 |
| Converted to Forest | 5.54 | 43.91 | 75.83 | 263.40 | 30.01 | 7.30 | 425.98 |
| Converted to Grassland | 29.25 | 803.62 | 156.46 | 3949.78 | 502.60 | 82.40 | 5524.11 |
| Converted to Unused land | 3.79 | 20.71 | 10.86 | 164.17 | 165.68 | 19.35 | 384.56 |
| Converted to Water area | 2.85 | 25.18 | 6.41 | 78.33 | 16.66 | 45.47 | 174.91 |
| 1980 Total | 71.84 | 1546.34 | 307.83 | 5675.41 | 819.99 | 214.16 | 8635.57 |
| Change | 790.27 | −282.44 | 118.15 | −151.3 | −435.43 | −39.25 | |

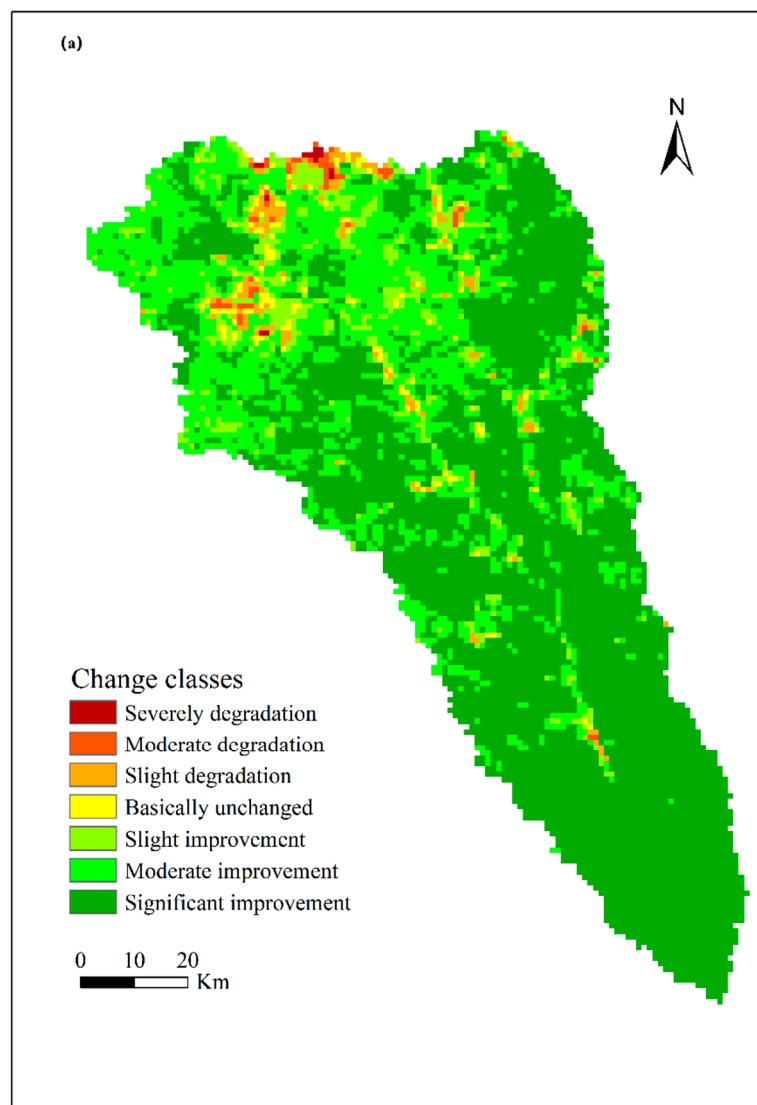

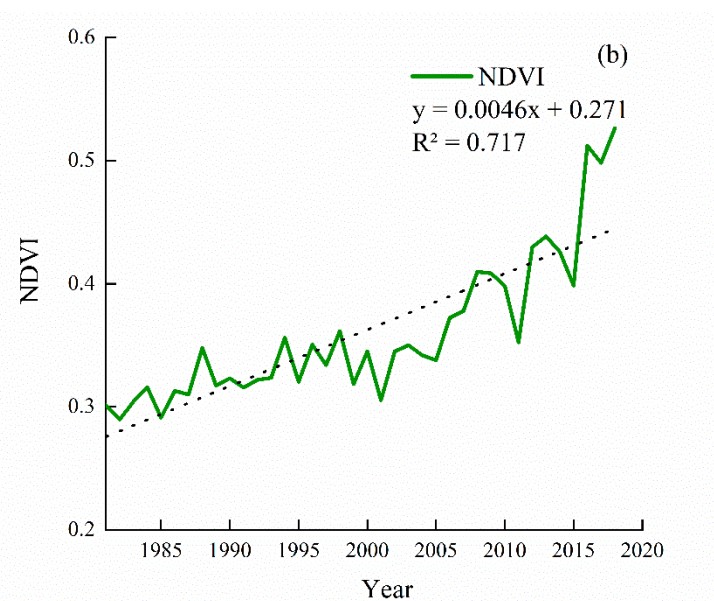

**Figure 8.** (**a**) Changes in the vegetation cover in the basin between 1980 and 2018 and (**b**) the normalized difference vegetation index (NDVI) trend fit.

*5.2. Conclusions*

In this study, trends in the precipitation, runoff depth, and potential evapotranspiration from 1956 2018 were analyzed using MK trend test methods. Four methods were also used to detect the change points in the runoff depth over the past 63 years. Then, based on the Budyko hypothesis, three single-parameter controlled Budyko methods were selected, and the elasticity coefficient method was used to analyze the changes in climate change (precipitation and potential evapotranspiration) and human activities (catchment surface characteristics). Finally, the contribution of climatic and anthropogenic factors to changes in the runoff in the Kuye River Basin was quantitatively evaluated, and the differences in the contribution of the three different Budyko methods to changes in the runoff and sensitivity of the influencing factors were compared and analyzed. The results showed that precipitation had a non-significant fluctuating upward trend, with a rate of 1.166 mm/decade, while the potential evapotranspiration had a decreasing trend, with a rate of 2.459 mm/decade. The runoff depth had an overall decreasing trend, with a rate of 1.159 mm/decade, which decreased significantly from 1996. The study period was divided into a base period, from 1956 to 1996, and anthropogenic impact period, from 1997 to 2018, based on the results of the four mutation tests. According to the results of the elasticity coefficients calculated via the three Budyko methods, during the study period, the runoff was more sensitive to changes in the precipitation, followed by the n parameter and potential evapotranspiration. For the precipitation elasticity coefficient, the sensitivity of the runoff depth from the Choudhury and Yang [38,39] equation was the highest during the base period while the sensitivity of the Wang & Tang [40] equation was higher in the anthropogenic impact period. Compared with the other two equations, the three elasticity coefficients from the Wang and Tang [40] equation fluctuated more. Within the range of relative errors, the relative contributions of the precipitation, potential evapotranspiration, and catchment surface characteristic parameters to the runoff were 1.30, −5.08, and −96.22% for the Wang and Tang [40] equation, respectively; the Fu [36,37] equation yielded relative contributions of precipitation, potential evapotranspiration, and catchment surface characteristic parameters to the runoff of 1.26, −3.63, and −97.64%, respectively, and the Choudhury and Yang [38,39] equation yielded relative contributions of precipitation, potential evapotranspiration and catchment characteristic parameters to the runoff of 1.28, −3.81, and −97.47%, respectively. Together, these three methods indicate that the influence of subsurface components is the most important factor for runoff variations in the Kuye River Basin.

**Author Contributions:** Methodology, J.Z.; software, J.Z.; validation, T.N.; formal analysis, Y.L.; investigation, J.Z.; resources, Y.H.; data curation, J.Z.; writing—original draft preparation, J.Z.; writing—review and editing, X.J.; visualization, Y.H.; supervision, Y.H.; project administration, Y.H. and X.J.; funding acquisition, Y.H. and X.J. All authors have read and agreed to the published version of the manuscript.

**Funding:** This research was supported by the Special Funds of the National Natural Science Foundation of China (Grant No.42041004), National Science Fund (51779209), and the Natural Science Basic Research Plan in Shaanxi Province of China (Grant No.2021JQ-449).

**Data Availability Statement:** The data presented in this study are available on request from the corresponding author. The data are not publicly available due to privacy.

**Conflicts of Interest:** The authors declare no conflict of interest.

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
