# Peer review of "Attribution Analysis of Runoff Variation in Kuye River Basin Based on Three Budyko Methods"

_land, doi:10.3390/land10101061_

Round 1
Reviewer 1 Report
The problem discussed in the article concerns the Budyka method as a tool for assessing the impact of climate change and human activity on changes in the outflow. The authors analyze trends using the observed data from 63 years on precipitation, depth of runoff and potential evapotranspiration. The authors derived three equations to describe the average annual runoff using the water balance equation and three Budyka formulas. Then, using the elasticity coefficient method, they quantified the contribution of climatic and anthropogenic factors in runoff changes in the Kuye basin for three Budyka models.
The research paper is interesting and scientifically sound. It is consistent with the scope of Land journal. The article is written clearly, the methodology used is correct, the inference is logical, and the results, apart from their cognitive values, may also be practically used for the selection of a method for determining the impact of climate and human activity on various environmental aspects.
However, there are some remarks concerning the basic assumptions and mathematical transformations:
- When determining ET0, knowledge of G - solid heat flux density is required, which is difficult to measure. Please complete section 2.2 Data sources with the description of the G evaluation.
- Based on tab. 1 and equation (6) it is impossible to derive this form of eq. (9) as presented by the authors. Probably the error is both in the equation (6) - incorrectly placed parenthesis, and in the equation (9). The reviewer could not verify the equation in tab. 1, based on the reference Fu [36, 37]. [36] is in Chinese and there is no such formula in [37]. If the presented results are based on the incorrect formula (9), then a significant part of the work will have to be recalculated.
Minor remarks:
Line |
Observation |
Solution |
137-175 |
Z (1– a/2), … |
correct a to α and explain what Z(1-α/2) means (percentile of the standard normal distribution) |
142 |
(mm day−1), |
(mm day−1), and consistently improve the indexes throughout the article |
197 |
3.2. Runoff Change Attribution Identification |
remove |
201 |
fpllows |
Correct to follows |
204 |
water storage change |
water storage (?) |
221 |
Budyko River |
Kuye River |
141 |
Eq. (1) Rn- G |
Rn - G |
256 |
Eq. (17) ??n? |
?n? |
377 |
96.22 %. |
- 96.22 %. (in tab. 4a) |
381 |
97.64 %, |
- 97.64 %, (in tab. 4b) |
385 |
59.81; 97% |
59.81 mm; 97.47% |
396 |
Tab. 4.c: 97.47% |
The value is positive what is in contradiction with the description in conclusions line 473 |
418 |
Table 7 1980-1018 |
Table 5 1980-2018 |
437 |
Table 5 |
Is it possible that all areas - construction land, cropland, forest grass-land and water have areas increased, nothing has decreased? |
Author Response
Response to Reviewer 1 Comments
Point 1: When determining ET0, knowledge of G - solid heat flux density is required, which is difficult to measure. Please complete section 2.2 Data sources with the description of the G evaluation.
Response 1: Thank you for your comments and your support for our work. We agree with your comment. About calculating ET0, First, basin-scale meteorological data were obtained based on GIS and then the results were derived using the official FAO tool for calculating potential evapotranspiration.
Point 2: Based on tab. 1 and equation (6) it is impossible to derive this form of eq. (9) as presented by the authors. Probably the error is both in the equation (6) - incorrectly placed parenthesis, and in the equation (9). The reviewer could not verify the equation in tab. 1, based on the reference Fu [36, 37]. [36] is in Chinese and there is no such formula in [37]. If the presented results are based on the incorrect formula (9), then a significant part of the work will have to be recalculated.
Response 2: Thank you for your comments and your support for our work. We agree with your comment. The â–³S (water storage change) in Eq.(6) can be considered as equal to 0 in the long time series. And we have fixed the error in the input of Table 1 and Eq. (9). In this situation, (9), (10), (11) can be deduced from the three equations in Table 1. combined with Eq. (6). In Table 1 we added the reference [24] in which the equation for Fu can be found. Our calculation is based on the correct formula, we have corrected the input error, thank you for your correction.
Point 3: Minor remarks
Response 3: Thank you for your comments and your support for our work. We have carefully revised the issues you pointed out, as detailed in the text. About Table 5, We added the changes in each type of land use from 1980-2018 and can see their overall increase or decrease.

Reviewer 2 Report
The manuscript titled as "Attribution Analysis of Runoff Variation in Kuye River Basin Based on Three Budyko Methods" by anonymous authors. The topic and results are well presented. The paper has little novelty; however, the topic may be of some interest.
I would recommend that the following criticisms be considered.
1) The authors investigated the trends in Kuye River watershed between 1956-2018 and divided the period as 1956-1996 and 1997-2018. What is the point to select this division? It should be clearly stated in the Methodology section.
2) It is better to move lines 94-109 to Study Area and Data section.
3) What are “UF”, “UB”, p, and data2 in Figure 3? The authors did not mention them in the text. Their explanation should be given under the figure title and in the text.
4) Missing Y-axis titles in Figures 5-8 should be given.
5) In line 398, “5. Conclusions and Discussion” should be “5. Discussion and Conclusions”.
Author Response
Response to Reviewer 2 Comments
Point 1: The authors investigated the trends in Kuye River watershed between 1956-2018 and divided the period as 1956-1996 and 1997-2018. What is the point to select this division? It should be clearly stated in the Methodology section.
Response 1: Thank you for your suggestions and your support for our work. We agree with your comment. Because of both climatic variability and human activities, the hydrological series has a significant variation in its statistical pattern around a certain point in time, which is defined as the variation point of the series. Analysis of runoff depth from 1956 to 2018 by the Pettitt test method, cumulative distance level method, moving t–test method, and Yamamoto method four mutation point test. The year 1997 was selected as the mutation year by combining various results. After finding the change points, the whole study period was divided into two periods: From 1956 to 1996 is the base period (assumed to be natural condition without human activities) and from 1997 to 2018 is the impacted period (human activities).
Point 2: It is better to move lines 94-109 to Study Area and Data section.
Response 2: Thank you for your comments and your support for our work. We decided to remove the figure and keep only the description of the phenomenon. Because we think this paragraph can lead to the research content and significance of this paper
Point 3: What are “UF”, “UB”, p, and data2 in Figure 2? The authors did not mention them in the text. Their explanation should be given under the figure title and in the text.
Response 3: Thank you for your comments and your support for our work. We agree with your comment. UF, UB, and p are the statistics in the Mann-Kendall trend test, respectively. UF is the standard normal distribution, calculated by time series, UB is the statistic calculated by inverse order of time series, and p refers to the confidence interval in the significance test. data2 is the error in making the graph has been modified at Figure 2.
Point 4: Missing Y-axis titles in Figures 4-7 should be given.
Response 4: Thank you for your comments and your support for our work. The Y-axis of Figure 4-7 has been given, see the text for details.
Point 5: In line 398, “5. Conclusions and Discussion” should be “5. Discussion and Conclusions”.
Response 5: Thank you for your comments and your support for our work. We agree with your comment. It has been changed to Discussion and Conclusions in line 398

Reviewer 3 Report
Before this manuscript can be accepted, following points needs to be revised:
- Abstract need to be rewritten to reflect the summary of this study.
- In the introduction section where you have mentioned about the objectives of this study, if previous paper already reported the effect of human activities on runoff, what is the point for this research? You need to rephrase it like we are here comparing effect of anthropogenic activities with climate change on runoff using three different methods.
- You need to explain why you have considered 1997-2018 as anthropogenic period, and how is different than that of base year?
- For other minor things, please check the attached reviewed manuscript

Author Response
Response to Reviewer 3 Comments
Point 1: Abstract need to be rewritten to reflect the summary of this study.

Response 1: Thank you for your comments and your support for our work. We agree with your comment. We have reworked the abstract to reflect the summary of this study. The Loess Plateau is the main soil erosion area within the Yellow River Basin; Quantifying the contribution rate of climate change and human activities to runoff change can provide support for water resources management in the Yellow River Basin. As a first-class tributary of the Yellow River, it was selected as the study area. Kuye River was selected as the study area. Runoff from the Kuye River Basin has decreased significantly since the 1990s owing to climate change and anthropogenic coal mining. The main objective of this study was to quantify the contribution and sensitivity of climate change and anthropogenic activities to runoff changes using three popular Budyko and elasticity coefficient methods, as well as to compare the similarities and differences among the three methods. The results show that: (1) Through four mutation point test methods, the change point of runoff in the study period of Kuye River Basin is 1997. (2) The elasticity coefficients calculated by the three Budyko methods showed that during the study period, the runoff was more sensitive to changes in the precipitation, followed by the catchment surface characteristic parameters and the potential evapotranspiration. (3) All three Budyko methods can yield a reasonable contributions of climate change and human activity to runoff changes. The three methods together indicate that the influence of the catchment surface characteristic parameters is the most important factor for the runoff variation in the Kuye River. For more details, please see the article
Point 2: In the introduction section where you have mentioned about the objectives of this study, if previous paper already reported the effect of human activities on runoff, what is the point for this research? You need to rephrase it like we are here comparing effect of anthropogenic activities with climate change on runoff using three different methods.
Response 2: Thank you for your comments and your support for our work. We agree with your comment. In previous studies, many scholars took only a single approach and had short time series. We have rephrased this statement and highlighted that this study uses a longer time series and three different methods to compare the effects of human activities and climate change on runoff.
Point 3: You need to explain why you have considered 1997-2018 as anthropogenic period, and how is different than that of base year?
Response 3: Thank you for your comments and your support for our work. We agree with your comment. We explain in section 4.2. Change Point Analysis of Runoff of the article.
We used 1997 as the change point of the runoff. This result is consistent with the timing of the massive coal mining in the Kuye River Basin and the implementation of the Chinese government's "return of farmland to forest" policy. This means that from 1997 to the present, a large number of anthropogenic activities have affected the natural ecological processes in the Kuye River Basin.
Point 4: For other minor things, please check the attached reviewed manuscript
Response 4: Thank you for your comments and your support for our work. We agree with your comment. We have rephrased the headings of Table 2 and Corrected other minor things. Thank you very much for your comments on this article.
